# Systemic *GFP* silencing is associated with high transgene expression in *Nicotiana benthamiana*

**Bill Hendrix**[1¤a]*, **Paul Hoffer**[1¤b], **Rick Sanders**[1], **Steve Schwartz**[1¤c], **Wei Zheng**[1¤d], **Brian Eads**[2], **Danielle Taylor**[2], **Jill Deikman**[1]

**1** Bayer Crop Science, Woodland, California, United States of America, **2** Bayer Crop Science, Chesterfield Parkway, St. Louis, Missouri, United States of America

¤a Current address: Bayer Crop Science, West Sacramento, California, United States of America
¤b Current address: Governor's Office of Emergency Services, Radiological Preparedness Unit, Mather, California, United States of America
¤c Current address: InnerPlant, Davis, California, United States of America
¤d Current address: Hangzhou Huadi Group Co., Hangzhou, China
* bill.hendrix@bayer.com

**Data Availability Statement:** All small RNA sequencing files are available in the SRA database under accession PRJNA695190. These data are the complete set of raw sequencing reads from the transgenic lines and F1 hybrid lines used in this

## Abstract

Gene silencing in plants using topical dsRNA is a new approach that has the potential to be a sustainable component of the agricultural production systems of the future. However, more research is needed to enable this technology as an economical and efficacious supplement to current crop protection practices. Systemic gene silencing is one key enabling aspect. The objective of this research was to better understand topically-induced, systemic transgene silencing in *Nicotiana benthamiana*. A previous report details sequencing of the integration site of the *Green Fluorescent Protein (GFP)* transgene in the well-known *N. benthamiana* GFP16C event. This investigation revealed an inadvertent co-integration of part of a bacterial transposase in this line. To determine the effect of this transgene configuration on systemic silencing, new *GFP* transgenic lines with or without the transposase sequences were produced. *GFP* expression levels in the 19 single-copy events and three hemizygous GFP16C lines produced for this study ranged from 50–72% of the homozygous GFP16C line. *GFP* expression was equivalent to GFP16C in a two-copy event. Local *GFP* silencing was observed in all transgenic and GFP16C hemizygous lines after topical application of carbon dot-based formulations containing a *GFP* targeting dsRNA. The GFP16C-like systemic silencing phenotype was only observed in the two-copy line. The partial transposase had no impact on transgene expression level, local *GFP* silencing, small RNA abundance and distribution, or systemic *GFP* silencing in the transgenic lines. We conclude that high transgene expression level is a key enabler of topically-induced, systemic transgene silencing in *N. benthamiana*.

study. The meta data includes all relevant information and references to figures in the manuscript to enable researchers to quickly cross-reference the raw data with the analyzed data we report. All other relevant data are located in the paper and its Supporting Information files.

**Funding:** Monsanto Company (now Bayer Crop Science) provided support in the form of salaries for all authors as well as logistical support for the studies. The funder reviewed and approved the manuscript for publication but did not play a direct role in the study design, data collection and analysis, or preparation of the manuscript. The specific roles of authors are detailed in the 'author contribution' section.

**Competing interests:** All authors were employed by Monsanto Company (now Bayer Crop Science) while conducting this research. This study was funded by Monsanto Company. Monsanto Company has filed a patent relevant to this study: ''Compositions and methods for delivery of a macromolecule or macromolecular complexes into a plant" (16/870,173). This does not alter our adherence to PLOS ONE policies on sharing data and materials.

## Introduction

RNA-based gene silencing is a sequence-specific, conserved mechanism in eukaryotes implicated in viral defense, control of transposable elements, and gene regulation. Gene silencing using transgenic approaches have been utilized to deploy a number of agriculturally important traits including virus resistance in papaya [1], delayed fruit ripening in tomato [2], black-spot bruise resistance and lower acrylamide levels post-cooking in potato [3], improved oil composition in soybeans [4] and insect control in corn [5]. These commercial products and others like them all take advantage of DCL-like proteins that cleave various forms of dsRNA into small interfering RNAs (siRNAs) 21-24nt in length. These siRNAs are loaded into an Argonaute protein and, along with other factors, form an RNA induced silencing complex, or RISC [6]. The RISC functions as a specific endonuclease that cleaves target transcripts identified by base pairing chemistry. Recently, methods have been developed to silence plant genes using topically delivered dsRNAs [7–11]. The most efficacious versions of these methods deliver 21-24nt dsRNAs that initiate silencing without initial dicer processing to produce efficacious gene silencing effectors. Topical dsRNA technology has the potential to complement and/or replace many of the current crop protection practices that are vital for agricultural productivity, but further research is needed to realize the potential of this new technology in agricultural production settings. Systemic gene silencing is one enabling aspect that needs more study. Systemic silencing of the *GFP* transgene of *N. benthamiana* line 16C has been reported using topical dsRNA delivery methods [7]. This study is part of an effort to understand how topically-delivered 22nt dsRNAs targeting *GFP* leads to a systemic silencing response in *N. benthamiana*.

*N. benthamiana* is model dicot that has been widely adopted by public and private sector researchers. This species is endemic to arid regions in Australia [12] and is noted for its virus susceptibility [13] and amenability to *Agrobacterium* infiltration [14]. Given these attributes, *N. benthamiana* is commonly used in studies examining plant-virus interactions and in studies utilizing the virus-induced gene silencing (VIGS) technique. Transgenic lines expressing a *Green Fluorescent Protein* (*GFP*) gene from *Aequorea victoria* were produced to study the initiation and maintenance of VIGS in *N. benthamiana* [15]. One of these transgenic lines, GFP16C, has since become a workhorse for research on many aspects of plant biology but most relevant to this report, local and systemic transgene silencing.

Systemic silencing of the *GFP* transgene in *N. benthamiana* was first reported by Voinnet and Baulcombe [14]. Local silencing was induced by *Agrobacterium* infiltration of a plasmid containing a T-DNA insert expressing the *GFP* coding sequence. Visual evidence of systemic *GFP* silencing was observed after infiltration as "unmasking" of red chlorophyll fluorescence along major and minor veins in distal, expanding tissue. The authors found no evidence that the bacteria or the T-DNA had migrated from the infiltration site and concluded the silencing signal originated in the infiltrated leaf but did not attempt to identify the mobile signal. The silencing signal in *N. benthamiana* is phloem-mobile, follows source-sink relationships [16], and can be impacted by light intensity [17]. In more recent work, researchers demonstrated that a DICER-LIKE2 (DCL2)-dependent mechanism is involved in the systemic spread of *GFP* silencing in *N. benthamiana* [18]. Using a grafting approach, the authors showed that *DCL2* was required in distal tissue to respond to mobile silencing signal but not required in the initiating tissue to produce the signal. In the same experiments, *DCL3* and *DCL4* were found to attenuate the systemic silencing response in *N. benthamiana*. Other reports in Arabidopsis show that *DCL2* is required in both rootstock and recipient shoot issue for efficient RDR6-dependent systemic silencing indicating mechanisms may differ across species [19].

Systemic silencing was also reported in tobacco [20]. Silenced transgenic lines expressing nitrate reductase, nitrite reductase, or glucuronidase were used in a series of grafting

experiments. The systemic silencing signal was transgene specific, unidirectional from stock to scion, and required a transcriptionally active transgene in the scion to propagate the silencing signal. Bidirectional systemic silencing has been reported in *N. benthamiana* [21, 22] and Arabidopsis [23]. A number of factors could contribute to differences in the observed patterns of silencing—including the model plant, the type of silencing (post-transcriptional versus transcriptional), the grafting method, and the developmental stage of the plant material [24].

Gene silencing can involve production of small RNAs for the targeted mRNA outside the dsRNA target region in both plant [25] and animal [26] systems. This phenomenon is referred to as transitivity. Small RNA transitivity is a feed-back loop that amplifies the initial silencing signal [27] and requires the action of an RNA-dependent RNA (RDR) polymerase. *RDR6* has been shown to be essential for transitive small RNA production in plants [28]. When *GFP* transgenes were targeted for silencing using a VIGS vector with a partial *GFP* coding sequence, abundant transitive small RNAs both 5' and 3' of the targeted sequence were observed in *N. benthamiana* (GFP16C) and Arabidopsis [29].

Transitivity is observed when targeting transgenes for silencing but reports of transitivity when targeting endogenous genes are mixed. Transitive small RNA production and systemic silencing for an endogenous gene, Virp1, and *GFP* were compared in *N. benthamiana* [30]. Systemic silencing and bidirectional transitivity were observed when silencing the *GFP* transgene but not the endogenous *Virp1* gene. Further, a *GFP* transgene with an endogenous gene promoter and intron did not exhibit transitivity, coding region methylation, or systemic silencing, but these molecular and phenotypic hall marks were observed when the *GFP* gene was driven by the CaMV 35S promoter and lacked an intron [27].

Another factor contributing to transitivity is the RNAi effector (trigger) that is used. Transitivity 3' of the target locus was observed using a 22nt amiRNA construct targeting chalcone synthase in Arabidopsis but not with a 21nt amiRNA construct [28]. These data indicate the occurrence of transitivity after silencing an endogenous gene may depend on the type of dsRNA used to initiate the silencing. Indeed, we have observed dsRNA-length dependent transitivity targeting both a *GFP* transgene and endogenous genes in our laboratories using *N. benthamiana*, tomato, and *Amaranthus cruentus* [9]. Production of transitive small RNAs may function to enhance local silencing and have been proposed to be essential for systemic silencing [24].

The gene integration site in the *N. benthamiana* GFP16C line was studied in detail [31]. A 3.2kb portion of a transposase gene from *Agrobacterium* was found co-integrated immediately adjacent to the *GFP* cassette. The authors suggested that the partial transposon may have an enhancing effect on the silencing response observed in the 16C line.

We conducted experiments in *N. benthamiana* to understand the impact of the partial transposase gene on local and systemic gene silencing and on transitive small RNA production after targeting the *GFP* transgene with a 22nt dsRNA delivered topically using carbon dot formulations. We found that the partial transposase had no impact on local silencing, systemic silencing, transitive small RNA production, or level of *GFP* expression. Using F1 hybrids of the 16C line in addition to a diverse set of new transgenic *GFP N. benthamiana* lines, we provide evidence that high *GFP* expression levels appear to be a major contributing factor to the sensitive systemic silencing response observed using the topical dsRNA technique in the GFP16C line.

## Materials and methods

### Plant growth conditions

All *N. benthamiana* plants were germinated in 200-cell plug trays prefilled with coconut coir plugs (Jiffy Preforma Blend 10) in a growth chamber maintained at 25°C with 150 μmol m$^{-2}$/s$^{-}$

[1] light intensity and a 16h day length. Relative humidity was not controlled and fluctuated according to irrigation frequency and plant density in the chambers at any given time. The seedlings were irrigated using an ebb and flow system 3 times per day with a dilute solution of 20-20-20 liquid fertilizer (Peters).

The seedlings were transplanted 9–10 days after seeding into 2.5in pots filled with Berger BM2 peat moss potting mixture. Transplants were grown with the same conditions described above except for irrigation frequency. Transplants were ebb-flow irrigated every other day for the first week and daily thereafter.

## Plant transformation

The T-DNA inserts for each transformation construct were synthesized using a third-party vendor (Bio Basic) based on sequences published by Philips et al. (2017). The inserts (S1 File) were cloned into a standard binary vector using *SpeI* and *NotI* restriction sites added during synthesis and sequence verified. *N. benthamiana* seedlings were transformed using *Agrobacterium tumefaciens* strain AB33 as described previously [32].

Regenerated shoots were transplanted as described above and sampled for *GFP* expression and copy number analysis using quantitative PCR. The single copy 16C line was used as a reference sample in these analyses. Seeds were harvested from the putative single-copy R0 lines. Forty R1 seeds per line were germinated in coir plugs and segregation of the *GFP* transgene was visually assessed to confirm the single copy designations made in the R0 generation. Putative homozygous seedlings were visually selected based on *GFP* fluorescence intensity. The selections were sampled for *GFP* copy number and expression analysis. Seeds were harvested from putative R1 homozygotes, and forty R2 seeds per line were grown to confirm the *GFP* locus was fixed in each line.

## DNA and RNA extraction and analysis

Leaf tissue was collected using a 4mm round biopsy punch. Eight to ten samples per leaf were collected into 96-well plate preloaded with steel grinding balls. The plates were frozen prior to sampling and tissue was collected on dry ice. Total RNA was extracted from leaf tissue using Trizol reagent (ThermoFisher). 1ml of Trizol was added to the frozen leaf discs. The plates were sealed, and the tissue was homogenized at 1600rpm for 10 min using a Genogrinder. The manufacturer's instructions were followed for the remainder of the procedure with exception of a 20 min centrifugation to precipitate total RNA. Glycol blue (45μg) was added to aid in pellet recovery. The RNA was resuspended in 20μl of RNase free water, and the concentration was measured using Quant-iT RNA BR assay kit (ThermoFisher). For qPCR analysis, the samples were diluted to 5ng/μl and target gene expression was measured as described previously [10]

DNA was extracted using Plant DNAzol (ThermoFisher) following the manufacturer's instructions. The purified DNA was resuspended in water and the concentration was measured using UV spectroscopy. The samples were diluted to 50 ng/μl and *GFP* copy number was estimated using qPCR. The qPCR reaction mixtures comprised DNA (100ng total), and the reactions were assembled as referenced for the expression analysis. Probes sets for NPTII and *GFP* coding region were utilized to estimate copy number relative to the 18S rRNA gene. The sequences for the all the primer and probes sets are provided in S1 Table.

## Topical dsRNA delivery

The dsRNA utilized in these studies were chemically synthesized and annealed by the manufacturer (Intergrated DNA Technologies). The dsRNA sequences are provided in S2 Table. All

dsRNAs were 22nt in length and contained 2nt 3' overhangs. The process used to select the efficacious dsRNAs was described previously [9, 33]. Briefly, Reynolds scores for all 19mers in a given coding sequences were calculated [34]. High scoring sequences were tested as 22nt dsRNAs for silencing efficacy measured by gene knock-down in protoplasts or gene knock-down and silenced area in whole plant testing. A minimum of 4 high-scoring sequences were screened for each gene.

Topical dsRNA delivery was performed using carbon dots produced in-house as described [10]. Briefly, the chemically synthesized dsRNAs were complexed to carbon dots overnight at room temperature in a solution containing 40 mM glycerol, 10 mM MES pH 5.7 and 12 μg/ml dsRNA. The dsRNA solution was added to solution of the same composition containing carbon dots. A carbon dot/ RNA ratio of 40–50 was utilized for all experiments. Prior to spray application the superspreading surfactant BreakThru S279 was added to the CD:dsRNA complexes at a final concentration of 0.4% (v/v). The solution was applied to the leaf surfaces using an Iwata HP-M1 handheld airbrush sprayer with air pressure set to 82 kPA (~12 PSI) held 2–3 cm away from the leaf surface. Approximately 60 μl of solution was applied to all leaves of each plant, in most cases a 3-4-leaf transgenic seedling. Whole plant images were collected 4–6 days after dsRNA to qualitatively assess *GFP* silencing. Plants were harvested and imaged for local and systemic *GFP* silencing 14 days after dsRNA application.

### Image capture and analysis

Leaves were harvested and placed on a black matte plastic imaging board. The leaves were photographed using an imaging station equipped with a Cannon EOS 70D camera with Canon lens (EFS 18-55mm lens, a low intensity white LED light source (EarthLED DirectLED™ 30271), and a high intensity LED royal blue light source (447 nm) model SL3500-D LED light panels with proprietary filters (Photon System Instruments). Images were acquired using the Cannon EOS utility 2 software with tethered image acquisition. For *GFP* images, 58mm Tiffen Green 11 and Yellow 12 filters were utilized to capture *GFP* and chlorophyll fluorescence from ~480nm to ~600nm.

The images were processed using ImageJ with the software provider's guidance. Briefly, the program operator utilized the threshold color panel to highlight a border around each leaf. A border image was overlaid onto the leaf image and the pixel number within the leaf border was quantitated by the software. The quantitated number of pixels represented the total leaf area. A similar thresholding process was used to highlight a border around the visible leaf phenotype and to quantitate pixels within the phenotypic area. *GFP* and *CHL-H* silenced areas were calculated by dividing the phenotypic area pixels by the total leaf area pixels.

### Small RNA library construction, sequencing, and analysis

Small RNA libraries were prepared using Illumina's TruSeq small RNA Library Preparation Kit according to the manufacturer's protocol (Document # 15004197v02) with modifications at the amplified cDNA gel purification step. Individual libraries with unique indexes were normalized by concentration and pooled by volume before gel purification. Pooled libraries were size separated with a 6% Novex TBE PAGE Gel and stained with 1X SYBR Gold for 20 minutes instead of ethidium bromide. Size selected libraries were sequenced using Illumina's NextSeq platform to a minimum depth of 3 million reads per sample.

Library quality was assessed using fastqc (Andrews, 2010) and Trimmomatic with read-quality filtering was used for trimming adapters [35]. For read mapping, processing and analysis, SAMtools [36], BAMtools [37], bowtie2 [38] and custom scripts (R and bash) were used. Counts of raw reads were normalized to the total number of reads passing length (18–48 nt)

and quality criteria (5 base sliding window with average quality above 20). Sequencing data files are available in the NCBI SRA database under Bioproject ID PRJNA695190.

## Statistical analysis

All data were analyzed using JMP Version 12 (SAS Institute Inc., Cary, NC).

## Results

### Generation and characterization of transgenic plants containing a *GFP* transgene with or without the 16C partial transposase element

To investigate the impact of the partial transposase sequence on systemic silencing in the 16C line, we synthesized T-DNA inserts containing either the full T-DNA sequence reported for the 16C line [31] or the same T-DNA sequence without the partial transposon. The T-DNAs were cloned into binary vectors and the cassette sequence was confirmed. pMON417669 comprised the insert including the selectable marker, the *GFP* expression cassette, and the partial transposase. pMON417670 comprised the same sequence without the partial transposase sequence (Fig 1A). Transgenic *N. benthamiana* plants were created with each construct. Ten single-copy events were selected in the R0 generation using qPCR to estimate copy number relative to the single-copy 16C line [15]. Copy number was confirmed in the R1 generation using transgene segregation and an additional round of qPCR copy number quantification. All events were confirmed as single copy in the R1 generation except event NT_W22241804

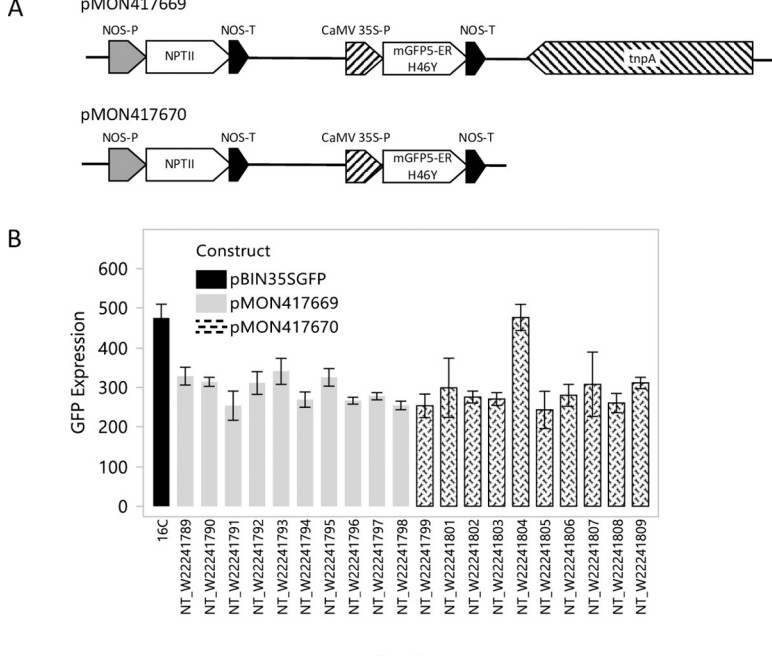

**Fig 1. *GFP* expression in 16C and homozygous transgenic lines with and without transposase.** Two binary vectors were constructed with T-DNA inserts comprising the 16C integration locus sequence described previously [31] (A) pMON417669 included a NPTII selectable maker, 35S:*GFP* expression cassette, and the partial transposase. pMON417670 included the same sequence without the partial transposase. (B). *GFP* expression for 16C and 20 transgenic events produced for this study. Tissue was collected from the first two true leaves of untreated seedlings. The data presented are from homozygous R2 lines. The experiment was arranged as a randomized complete block with four replications per event. *GFP* expression values are calculated relative to the *PP2a* gene. The data are expressed as means +/- standard error.

(pMON417670) which was an unlinked, two-copy event. Homozygosity was confirmed in the R2 generation for all events prior to utilization of the lines for experimentation.

*Green fluorescent protein* (*GFP*) expression was measured in leaves of four homozygous plants per event using qPCR in the R2 generation. *GFP* expression values ranged from 51–72% of the 16C line in single copy events (Fig 1B). The two-copy NT_W22241804 event had *GFP* expression equivalent to the 16C line. The same plants sampled for expression were utilized in the first repetition of the systemic silencing screening experiment.

## Topical dsRNA delivery and target gene silencing in transgenic *GFP* events

Short dsRNAs 22nt in length were chemically synthesized and used to target the *GFP* and *magnesium chelatase subunit H* (*CHL-H*) genes in the 16C line (S2 Table). Carbon dot formulations were used to topically deliver these dsRNAs or a scrambled control sequence to 16C seedlings [10]. Application leaves from the plants were removed and photographed 6 days after dsRNA application (Fig 2). Visual indications of gene silencing were evident for *GFP* and *CHL-H*. *GFP* silencing appeared as red chlorophyll fluorescence on application leaves against the green fluorescent background when the leaves were excited with a blue light source (Fig 2A top). *CHL-H* silencing appeared as yellow sectors on the application leaves (Fig 2B top). Tissue was collected from phenotypic areas to measure gene expression and small RNA abundance. Reduced mRNA levels were observed for both *GFP* (Fig 2A middle) and *CHL-H* (Fig 2B middle) when those genes were targeted by a specific dsRNA. Transitive small RNAs were observed both 5' and 3' of the target region for the *GFP* gene (Fig 2A bottom), but only 3' of the target region for *CHL-H* gene (Fig 2B bottom). The transitive small RNAs were predominantly 21nt in length but other biologically important size classes (e.g. 22nt and 24nt) were also observed (Fig 2 inset bottom).

The *GFP* transgene was silenced in the 16C line and the 20 transgenic events produced for this study using carbon dot delivery of a chemically synthesized 22nt dsRNA targeting the *GFP* transgene. Whole plants were photographed 4 days after dsRNA application to qualitatively assess local *GFP* silencing (Fig 3A top). The plants were harvested by removing all the leaves 14 days after dsRNA application. The leaves were arranged in developmental order and photographed under blue light (Fig 3A bottom). These images were analyzed for local *GFP* silencing on the application leaves (Fig 3B top) and for systemic silencing on younger leaves (Fig 3B bottom). Systemic *GFP* silencing covering 25% and 12% of the total leaf area was observed in the 16C and NT_W22241804 lines, respectively. Weak systemic *GFP* silencing was observed in many of the other events, but the silenced area was low, and did not continue to spread into new tissue like in the 16C and NT_W22241804 lines. In most instances, the systemic silencing in these events was visually evident in only 1 or 2 leaves, many times appearing in single or a few veins. No difference was observed in the extent or frequency of systemic silencing comparing the events containing the partial transposase and the events without the partial transposase. The experiment was conducted three times with similar results. The data from the experiment with highest local *GFP* silencing is shown.

## Silencing in hemizygous 16C lines

Given our inability to reproduce the high expression levels observed in the 16C line or 16C-like systemic silencing in any single-copy transgenic line, we wanted to better understand the role of *GFP* expression levels in systemic transgene silencing. To do this, we examined the systemic silencing response in three F1 lines hemizygous for the 16C event. Each line originated from an independent cross of the 16C line as a male parent and three different wildtype *N. benthamiana* plants as the female parents. As expected *GFP* expression was reduced by

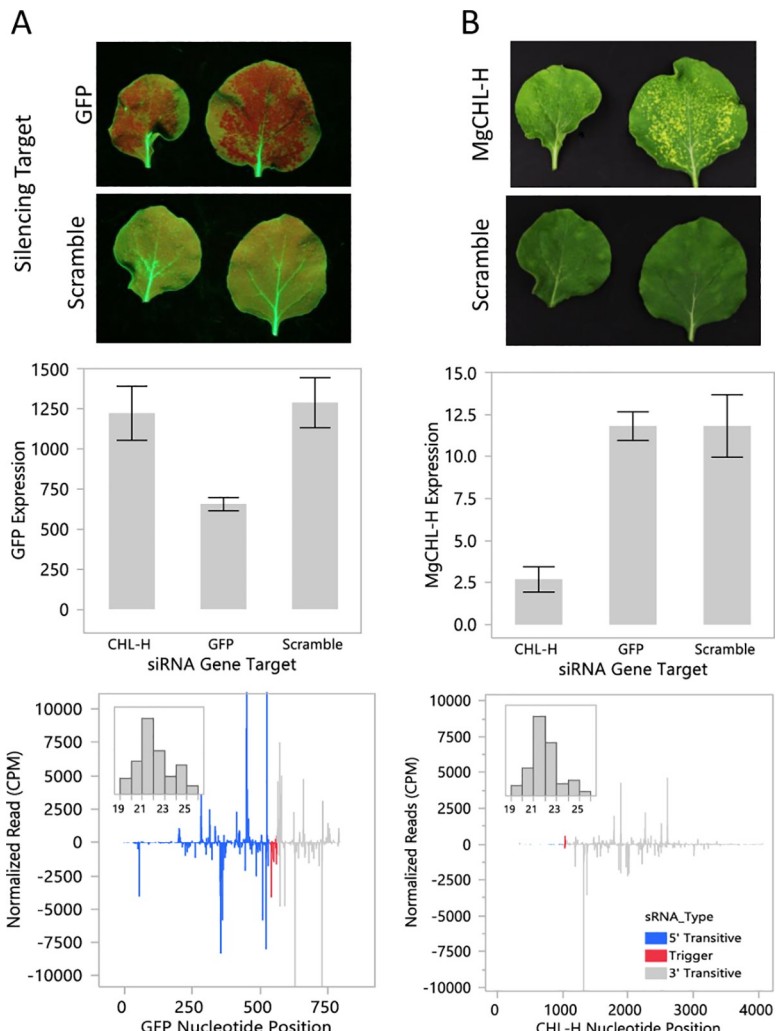

**Fig 2. Topical dsRNA delivery using carbon dots.** Short dsRNAs 22nt in length were delivered topically to *N. benthamiana* using carbon dot technology. The *GFP* transgene and the *magnesium chelatase subunit H* (*CHL-H*) were targeted in the 16C line. Application leaves were harvested 6 days after dsRNA treatment. Visual phenotypes were observed for *GFP* (A, top) and *CHL-H* (B, top). Target gene expression and small RNA production were measured in tissue collected from phenotypic leaf sectors and non-phenotypic control tissues. *GFP* (A, middle) and *CHL-H* (B, middle) expression was reduced 48 and 72%, respectively. Abundant transitive small RNAs were observed both 5' and 3' of the target region for *GFP* (A, bottom). Transitive small RNAs were only observed 3' of the target region for *CHL-H* (B, bottom). The experiment was arranged as a randomized complete block with 4 replications of each treatment. The expression data are expressed as means +/- standard error calculated relative to the *PP2a* gene. The replicates for each treatment were pooled prior to small RNA sequencing. The sequencing data are expressed as the sum of normalized small RNA counts per $1\times10^6$ reads for RNAs 19-25nt in length with positive and negative values represent sense and antisense reads, respectively.

approximately half in the hemizygous F1 lines (Fig 4B inset). Local *GFP* silencing was induced using carbon-dot delivery of a 22nt dsRNA targeting *GFP* with a single application of the formulation or two applications of the formulation 4 days apart. The plants were harvested, photographed and sampled for small RNA sequencing. 16C-like systemic *GFP* silencing was not observed in any of the hemizygous 16C lines (Fig 4A). Minor vein silencing was observed in 1 or 2 leaves in some of the hemizygous plants. In these cases, the observed *GFP* systemic silenced area was reduced more than 100-fold relative to the 16C homozygous control (Fig 4B).

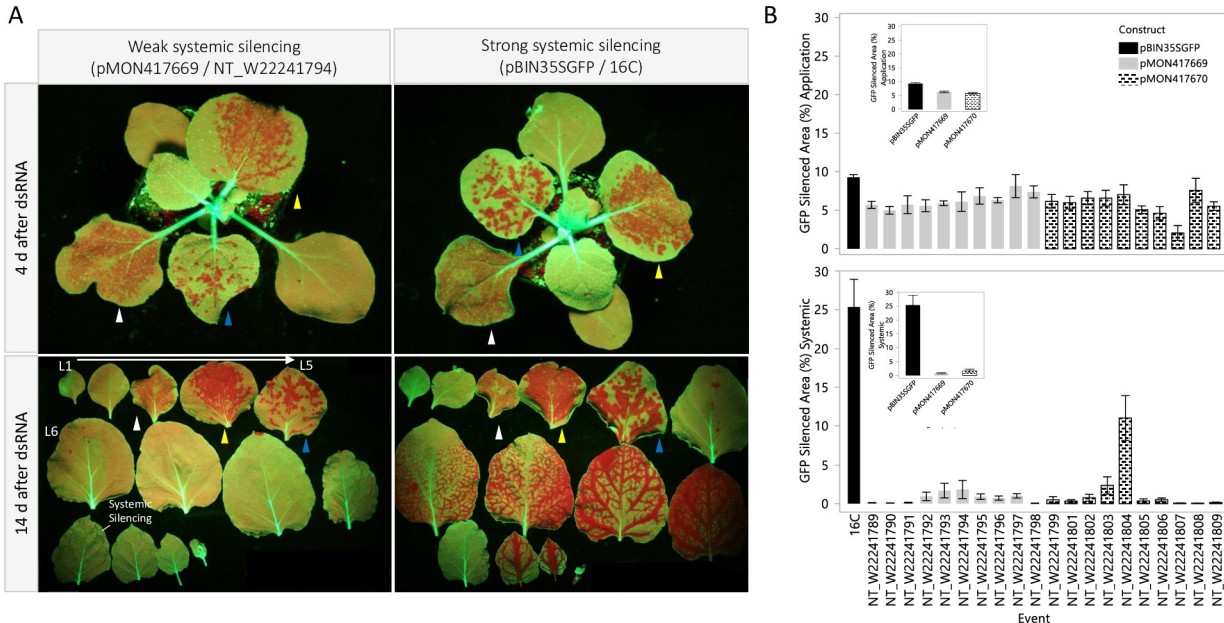

**Fig 3. Local and systemic *GFP* silencing in 16C and homozygous transgenic lines with and without the partial transposase.** A 22nt dsRNA targeting *GFP* was topically delivered to transgenic R2 *N. benthamiana* seedlings homozygous for the *GFP* locus using carbon dots. Intact plants were photographed at 4 d after dsRNA treatment to qualitatively assess *GFP* silencing (A, top). The plants were destructively harvested 14 d after dsRNA treatment. All leaves were removed, arrayed in developmental order, and photographed (A, bottom). Application leaf identities are denoted by the colored arrows. Local (B, top) and systemic (B, bottom) *GFP* silenced area was measured using ImageJ. Developmental abnormalities and extreme stunting were observed for event NT_W22241807. Systemic *GFP* silencing covering 25% and 12% of the total leaf area was observed for the 16C line and the two-copy NT_W22214804 event, respectively. Low levels of systemic *GFP* silencing were observed in the remaining events. The partial transposase had no impact on local or systemic *GFP* silencing (inset) 14 days after dsRNA treatment. The experiment was conducted 1 time in the R1 generation and 2 times in the R2 generation. Each repetition was arranged as a randomized complete block with 4 replications per treatment. The phenotypic data are means for 4 replicates +/- standard error from the experiment with the greatest local *GFP* phenotypes and levels of systemic *GFP* silencing.

## Transitive small RNA production across transgenic and hemizygous 16C lines after *GFP* silencing

Small RNA profiles were measured for application and systemic leaves from tissue collected 14 days after dsRNA application for all the lines in the systemic silencing screen. For application leaves, tissue was collected from sectors with a visible *GFP* silencing phenotype. For systemic leaves, tissue was collected from phenotypic areas where possible. In the absence of a visible systemic silencing phenotype, tissue was collected from the midrib and surrounding tissue from an expanding leaf 2–3 leaves away from the apex of the plant. Abundant transitive small RNAs targeting the coding region of the *GFP* gene were observed in application leaves in the 16C line (Fig 5A). These small RNAs were distributed both 5' and 3' of the targeted region of the *GFP* gene (Fig 5B) and well above the background small RNA levels observed in untreated tissue for all events (S1 Fig). Similarly, both 5' and 3' transitive small RNAs were observed in application leaves for all the transgenic events generated for this study and in the 16C hemizygous lines. The transitive small RNAs in the application leaves were, on average, 10-fold less abundant in the events created for this study relative to the 16C line (Fig 5A). Somewhat higher levels of transitive small RNAs were observed in the application leaves of the hemizygous lines. The F1-7 line had 3' transitive small RNA quantities approximately equal to the counts observed for the homozygous 16C line.

Small RNAs mapping to the *GFP* transgene were observed in the systemic leaves of all events. Generally, the abundance was low and near background for most lines (S1 Fig).

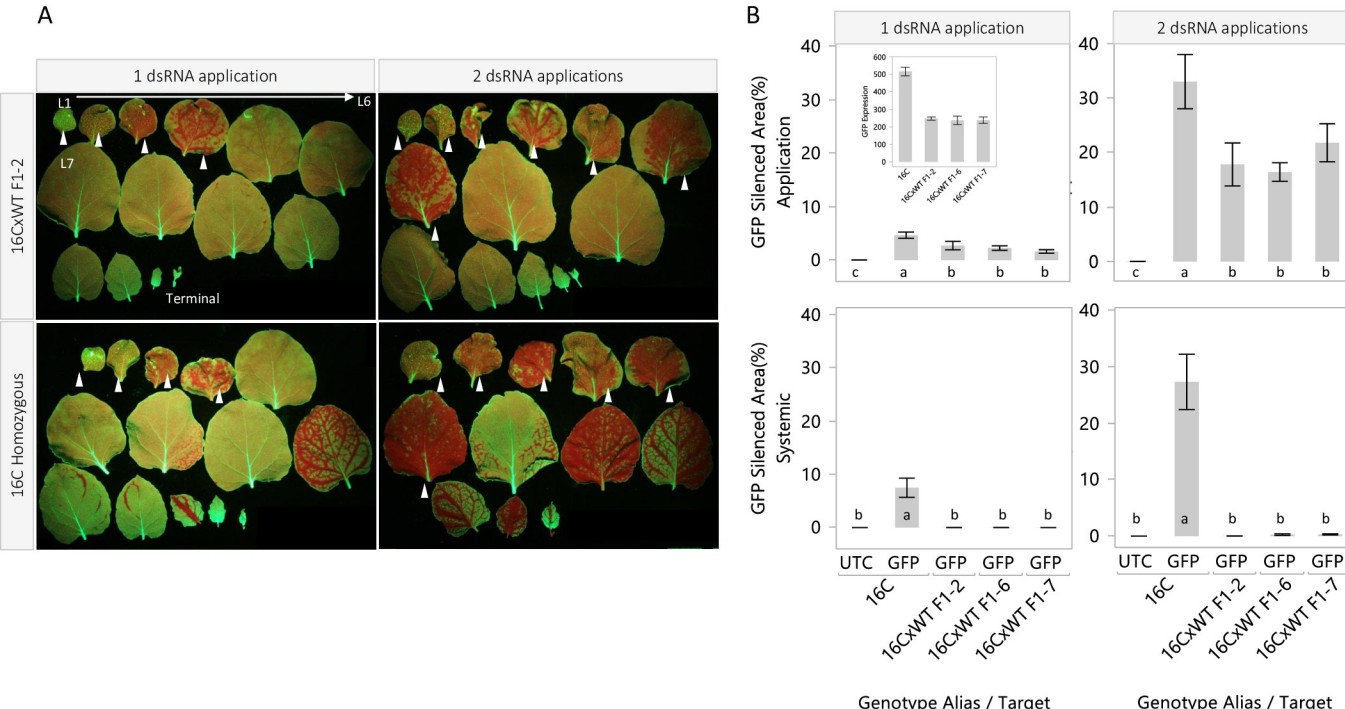

**Fig 4. Local and systemic silencing in 16C hemizygous lines.** The systemic *GFP* silencing response was evaluated in three 16C hemizygous lines. The seedlings were topically treated with one or two applications of dsRNA/carbon dots solution. The plants were harvested 14 d after dsRNA treatment. All leaves were removed, arrayed in developmental order, and photographed. The white arrows denote the application leaves. (A) *GFP* expression was measured using qPCR. *GFP* was reduced by approximately half in the hemizygous lines relative to 16C (B, inset). *GFP* silencing in the application and systemic leaves was measured using Image J. *GFP* silencing was observed in application leaves for all treated plants. However, the silenced area was significantly reduced in the hemizygous lines relative to the 16C homozygous control (B, top). Systemic *GFP* silencing was observed in the 16C homozygous line covering 7.5 and 27.2% of the total leaf area in the single and double application treatments, respectively (B, bottom). The levels of systemic silencing in the hemizygous plants were low and not significantly different from the untreated 16C control. The experiment was arranged as a randomized complete block with 4 replications per treatment. The data are means +/- standard error. Letters indicate statistical difference using Student's t-test (α = 0.05).

Abundant systemic small RNAs were observed for both 16C and the two-copy NT_W22241804 lines. Similar to application leaves, the systemic small RNA abundance in the 16C line was up to 2 orders of magnitude higher than observed for any comparator line. The 21nt transitive small RNAs were the most abundant small RNA size class in both application and systemic leaves (Fig 5B inset). Total small RNA counts in application leaves were weakly correlated to small RNA counts in systemic leaves (Fig 5C).

## Discussion

The *N. benthamiana GFP* reporter line 16C has been used extensively to study plant-virus interactions, transgene silencing and many other areas of plant biology. In our early topical gene silencing experiments using the 16C line, we observed local *GFP* silencing and in many cases systemic *GFP* silencing 7–14 days after topical dsRNA application. With further study, we learned that the systemic *GFP* silencing in 16C could be specifically initiated using 22nt dsRNA [7, 9] and that topical delivery of dsRNA targeting *GFP* in the 16C line initiated an amplification process that is characterized by production of transitive small RNAs both 5' and 3' of region targeted with dsRNA, especially when using a 22nt dsRNA [9].We adapted the topical dsRNA technology to several other dicot species targeting both endogenous genes and transgenes. However, we were unable to identify another genetic system in which we observed systemic gene silencing after topical dsRNA application [10, 33]. Further, the observation of

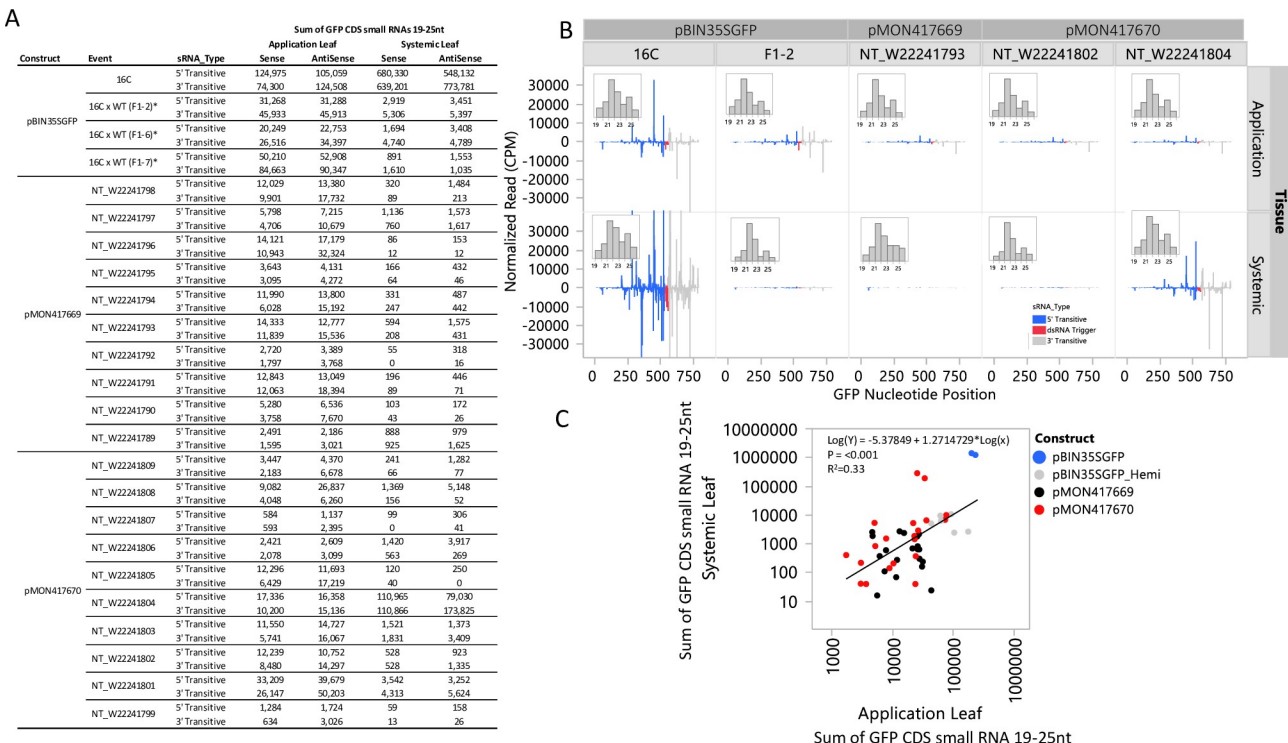

**Fig 5. Transitive small RNA production 14 d after dsRNA application in local and systemic tissues.** The small RNAs from phenotypic application and systemic leaves sampled 14 days after dsRNA application were sequenced and mapped to the coding sequence of the *GFP* transgene. Transitive small RNAs were observed both 5' and 3' of the target region for all application leaf samples evaluated (A). Substantial variation spanning two orders of magnitude was observed for the total number of small RNAs mapped in these samples. In systemic tissue, 5' and 3' transitive small RNAs were observed for the 16C and NT_W22241804 line. Consistent with the visual phenotypic difference (Fig 3), the 16C line had 10-fold more total small RNAs than observed in the NT_W22241804 line. The other systemic samples had small RNAs at or near background levels (S1 Fig). The transitive small RNAs were distributed across the entire length of the *GFP* transgene and were predominantly 21nt in length (B). A weak correlation ($R^2$ = 0.33) was observed between application leaf transitive small RNAs and systemic leaf transitive small RNAs (C). The replicates for each treatment were pooled prior to small RNA sequencing. The sequencing data are expressed as the sum of small RNA counts 19-25nt in length per $1 \times 10^6$ total small RNA reads with positive and negative values representing sense and antisense reads, respectively.

abundant transitive small RNAs 5' of region targeted was not replicated in most transgenes and all endogenous gene targets that we were able to silence using a topical dsRNA technique.

A partial transposase sequence was integrated immediately adjacent to the *GFP* expression cassette in the 16C line [31]. Since transposable elements are known targets of gene silencing pathways in plants [39, 40] it seemed possible that the proximity of such a transposase sequence to the *GFP* transgene could affect silencing activity at that locus, we completed experiments to test if the partial transposase sequence in the 16C line facilitated and/or enhanced systemic transgene silencing. We utilized the published sequence to synthesize T-DNA inserts containing the same expression elements, including repeated elements such as the NOS terminator, and any deviation from the originally published 16C T-DNA sequence with and without the partial transposase sequence (Fig 1). We did not observe any enhancing effect on systemic silencing as a result of including the transposase sequence in the transformation constructs (Fig 3). We also didn't observe any effect on the level of expression of the *GFP* transgene (Fig 1).

We observed both 5' and 3' transitivity for the *GFP* transgene after topical application of a 22 bp dsRNA but only 3' transitivity was observed for *CHL-H* (Fig 2). It is unclear what drives the production of sRNAs 5' of the targeting site in the case of the *GFP* transgene, but high expression level may play a role. In the case of *GFP* versus *CHL-H* in these experiments, GFP is expressed approximately 100-fold higher than *CHL-H*. We have observed transitivity in the 3'-

only direction for CHL-H in *Amaranthus cruentus* [9] and other endogenous genes not discussed here. Given these observations, we considered that 5' transitivity could be unique to the 16C event and perhaps associated with systemic silencing. We were able to replicate the 5' transitivity phenomenon for the *GFP* transgene in all of the transgenic events produced for this study (Fig 5). However, 5' transitivity was not predictive of systemic silencing in these lines.

High expression level and, to a lesser extent, small RNA abundance in application leaves were the only molecular parameters associated with systemic silencing in the new transgenic events. We achieved *GFP* expression equivalent to the 16C line in a two-copy pMON417670 event. The integration loci were not linked based on segregation data, but the locus arrangement and any spurious integrations in this line were not investigated. Given this uncertainty, we also used F1 hybrid lines of 16C crossed with wildtype *N. benthamiana* to ascertain if systemic silencing was observed when expression from the 16C locus was reduced, in these cases by roughly half. Systemic silencing initiated with topical dsRNA was minimal in these hybrid lines. These data taken together suggest high expression is a key feature that enables the robust systemic silencing in the 16C line when initiated using carbon dot/dsRNA formulations. However, given the equivalent *GFP* expression and somewhat attenuated systemic silencing response in the two-copy line NT_W22241804, other factors such as integration locus effects may contribute to the more robust systemic response in the 16C line.

Our data provide support to the tiered threshold model explaining spontaneous *GFP* silencing in *N. benthamiana* proposed previously [41]. In this model, the authors propose that cellular dsRNA and mRNA levels are both involved in progression from an initial silencing event (transcript cleavage) to local silencing and then on to systemic silencing. Our results suggest mRNA expression level is more impactful than local dsRNA levels using the topical, carbon-dot delivery system. Local *GFP* silenced area was increased 4-6x (Fig 4) when using two dsRNA applications in the experiments examining systemic silencing in the 16C hemizygous lines. We did observe a small increase in systemic silencing in the hemizygous lines, but the levels did not approach the increase in systemic silencing observed in the 16C line when using two dsRNA applications. Further, transitive small RNA counts from phenotypic application leaves were only weakly correlated to transitive small RNA counts in the systemic tissue, explaining only 33% of the variation in the systemic samples (Fig 5C). These results taken together suggest that increasing the initial silencing "burst" using the topical, carbon-dot delivery system is not enough to induce 16C-like systemic silencing in transgenic lines with *GFP* expression that is 28–50% lower than 16C. We did not investigate the impact of agroinfiltration or other efficient silencing inducer systems on the development of systemic silencing in these lines. These experiments could provide further data on the relationship of the strength and duration of the local silencing induction, expression level, and systemic silencing. The formation of aberrant RNAs as a result of high transgene gene expression [42] may be another factor contributing to systemic silencing in 16C and the two-copy line, but further study is needed to understand the role aberrant transcripts may play in systemic silencing in these lines.

We investigated the systemic *GFP* silencing response in the widely used *N. benthamiana* transgenic line, 16C. We were unable to replicate the systemic response in a single copy line, but we were able to rule out the co-integrated bacterial transposase as an enabling genetic component when initiating silencing using the topical dsRNA technology developed at Bayer. Further, transitive small RNA production 5' of the *GFP* target region was not predictive or enabling of systemic transgene silencing. We conclude high transgene expression level is an important enabling factor for self-sustaining, systemic gene silencing using the topical dsRNA technology described it this report.

## Supporting information

**S1 Fig. Background levels of small RNAs mapping to the *GFP* gene in untreated *N. benthamiana*.** Untreated tissue from the first true leaf of each homozygous R2 transgenic events was sampled and the small RNAs were sequenced. The experiment was arranged as a randomized complete block with 4 replications per treatment. The replicates for each treatment were pooled prior to small RNA sequencing. The sequencing data are expressed as the sum of small RNA counts 19-25nt in length per $1 \times 10^6$ total small RNA reads.
(TIF)

**S1 Table. Primer and probes sequences used for qRT-PCR in this study.**
(DOCX)

**S2 Table. dsRNA sequences utilized in this study.**
(DOCX)

**S1 File. T-DNA sequences of the transformation constructs utilized in this study.**
(PDF)

## Acknowledgments

We thank Professor David Baulcombe for sharing the N. benthamiana 16C GFP line. We thank Jim Byrne, Brenda Reed, and Kaylene Yandel for their assistance in growing and maintaining the plants used in this study. We thank Ericka Havecker for discussions and helpful feedback on these studies.

## Author Contributions

**Conceptualization:** Bill Hendrix, Steve Schwartz, Wei Zheng, Jill Deikman.

**Data curation:** Bill Hendrix, Paul Hoffer, Rick Sanders, Brian Eads, Danielle Taylor.

**Formal analysis:** Bill Hendrix, Paul Hoffer, Brian Eads.

**Investigation:** Bill Hendrix, Paul Hoffer, Rick Sanders, Steve Schwartz, Danielle Taylor.

**Methodology:** Bill Hendrix, Paul Hoffer, Rick Sanders, Steve Schwartz, Wei Zheng.

**Project administration:** Bill Hendrix, Jill Deikman.

**Software:** Paul Hoffer, Brian Eads.

**Supervision:** Bill Hendrix, Jill Deikman.

**Validation:** Bill Hendrix, Paul Hoffer, Rick Sanders.

**Visualization:** Bill Hendrix, Paul Hoffer.

**Writing – original draft:** Bill Hendrix.

**Writing – review & editing:** Paul Hoffer, Rick Sanders, Steve Schwartz, Wei Zheng, Brian Eads, Danielle Taylor, Jill Deikman.

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
