## [Decision Letter · Decision Letter 0]

20 Jan 2021

PONE-D-20-40871

High transgene expression is associated with systemic GFP silencing in Nicotiana benthamiana

PLOS ONE

Dear Dr. Hendrix,

Thank you for submitting your manuscript to PLOS ONE. After careful consideration, we feel that it has merit but does not fully meet PLOS ONE’s publication criteria as it currently stands. Therefore, we invite you to submit a revised version of the manuscript that addresses the points raised during the review process.

The current manuscript does not fully meet the publication criteria of the conclusions being supported by the provided results and the methods being adequately described. Note that the decision of Major Revision is based on assumption that some additional data will need to be provided to address Reviewer 1's concerns.  Both reviewers have provided suggestions that would generally improve the manuscript that should also be considered.

We look forward to receiving your revised manuscript.

Kind regards,

Keith R. Davis

Academic Editor

PLOS ONE

Journal Requirements:

We note that one or more of the authors are employed by a commercial company: Bayer Crop Science, InnerPlant, Hangzhou Huadi Group Co.

2.1. Please provide an amended Funding Statement declaring this commercial affiliation, as well as a statement regarding the Role of Funders in your study. If the funding organization did not play a role in the study design, data collection and analysis, decision to publish, or preparation of the manuscript and only provided financial support in the form of authors' salaries and/or research materials, please review your statements relating to the author contributions, and ensure you have specifically and accurately indicated the role(s) that these authors had in your study. You can update author roles in the Author Contributions section of the online submission form.

2.2. Please also provide an updated Competing Interests Statement declaring this commercial affiliation along with any other relevant declarations relating to employment, consultancy, patents, products in development, or marketed products, etc.  

Reviewers' comments:

Reviewer's Responses to Questions

**Comments to the Author**

1. Is the manuscript technically sound, and do the data support the conclusions?

Reviewer #1: Partly

Reviewer #2: Yes

2. Has the statistical analysis been performed appropriately and rigorously? 

Reviewer #1: Yes

Reviewer #2: Yes

3. Have the authors made all data underlying the findings in their manuscript fully available?

Reviewer #1: Yes

Reviewer #2: Yes

4. Is the manuscript presented in an intelligible fashion and written in standard English?

Reviewer #1: Yes

Reviewer #2: Yes

5. Review Comments to the Author

Reviewer #1: The authors produced a series of N. benthamiana lines expressing GFP transgene constructs. Single copy events were used to compare their response to a silencing trigger with seedlings of the 16C line. As a transgene the newly generated lines contained either the authentic genome-integrated transgene sequence of the 16C line (pMON417669) or they harboured the 16C line sequence lacking the partial transposase (pMON417670). GFP expression analysis revealed no significant differences in GFP expression levels between individuals of the two lines indicating that the transposase sequence did not enhance GFP expression. In contrast, a transgenic line (NT_W22241804) containing two copies of the pMON417670 transgene displayed a GFP expression level that was similar to that of the 16C line.

Using a topically applied synthetic 22nt GFP-targeting siRNA, local silencing was induced in all plants with strong silencing phenotypes in the 16C and NT_W22241804 lines and weak GFP silencing in all other lines. sRNA-seq data demonstrated that secondary sRNA production was initiated in all locally silenced lines. 22nt siRNA-mediated silencing of an endogenous CHL-H gene was also associated with transitivity but the secondary sRNA profile clearly differed from the GFP-specific sRNA profiles. Importantly, initiation of systemic silencing was only found upon local silencing of the 16C and NT_W22241804 lines. Based on these findings together with the observation that GFP hemizygous progeny of genetic cross of the 16C line with N. benthamiana wildtype plants failed to develop systemic silencing post 22nt siRNA application, the authors concluded that induction of systemic silencing is correlated with high levels of transgene expression.

Major points:

We conclude that high transgene expression level is a key enabler of systemic transgene silencing in N. benthamiana.

In addition to the suggested high transgene expression level, the efficiency of the silencing trigger may have an impact on the initiation of systemic silencing. However, this issue has not been addressed in the present study. Thus, the conclusion that the transgene expression level is a key enabler of systemic transgene silencing should be taken with care. Although the CD-associated 22nt siRNA induced systemic silencing in the homozygous 16C line application of CD:dsRNAs may have an impact on the initiation of systemic silencing. Thus, this reviewer strongly recommends to conduct agroinfiltration experiments using hemizygous 16C plants. Infiltration of an efficient GFP silencing inducer, e.g. GFP hairpin RNA transgene construct, will provide additional evidence that hemizygous 16C plants are indeed unable to develop systemic silencing.

sRNA profiles: sense and antisense of 21, 22 an 24nt siRNAs should be presented. The profiles of 19-25nt sRNAs may be included in the Supporting Information. Sense sRNA profiles also contain GFP degradation products. Thus, it is important to separately analyse sense and antisense sRNA reads. A focus on 21, 22 and 24nt siRNAs will accentuate the number and distribution of the putatively functional sRNAs that are predominantly DCL products.

In particular, in the “systemic leaves” of NT_W22241793 and NT_W22241802 lines, only the presence of antisense sRNAs will indicate that the accumulating sRNAs derived from the silencing machinery. In this context, it is also important to show CHL-H-specific antisense siRNA (21, 22, ant 24nt) accumulation.

The authors should also comment on the observation that “5’ transitive” sRNAs of the CHL-H transcripts are missing. Why no 5’ mRNA degradation products with sense orientation are detected? Finally, in Fig. S1, separate presentation of sense and antisense sRNAs will probably demonstrate that the sRNAs only correspond to GFP-specific degradation products (only sense sRNAs) and not to DCL products. This reviewer predicts that the number of antisense sRNAs, if detectable at all, will be extremely low. In Fig. S1, the size distribution of the sRNAs would make sense. It will show if 21nt sRNAs are predominantly accumulating or if 19 to 25 nt sRNAs are almost equally accumulating. Equal distribution of 19 to 25 nt sRNAs would argue for a mRNA degradation-based origin.

Figure 2S is missing.

Minor points:

Page 11, line 86: Using a grafting approach, the authors showed that DCL2 was required in distal tissue to respond to mobile silencing signal but not required in the initiating tissue to produce the signal. This issue is controversially discussed. Taochy et al. (2017) reported that DCL2 is required in both the source rootstock and the recipient shoot tissue for efficient RDR6-dependent systemic PTGS.

Page 15, line 192: The solution was applied to the leaf surfaces using an Iwata HP-M1 handheld airbrush sprayer. Please, indicate the applied pressure.

Where the authors purchase from Silwet S279? I only know BreakThru S279 and Silwet L-77.

References should be revised. See Chen et al., 2018, name of the journal (Plant Physiol) is missing or McHale et al. 2013, The Plant journal : for cell and molecular biology. 534 2013;76(3):519-29.

Please, note that References were not fully reviewed!

Reviewer #2: The manuscript of Bill Hendrix and colleagues titled "High transgene expression is associated with systemic GFP silencing in Nicotiana benthamiana" presents results of the study focused on the reasons that predispose some transgenic lines to systemic silencing after topical application of dsRNA. The impact of three possible factors was assessed; i) presence of transposase gene fragment, ii) occurrence of 5’ transitivity and iii) transgene expression level. The study is smartly designed, well described and the results are well interpreted. Based on the results, the authors conclude that the high expression level is associated with systemic silencing, while neither of the other two factors seemed to be involved. Since the main conclusion is based just on two transgenic lines, the authors admit that the impact of some other (co-)factors cannot be excluded. However, since the hemizygous lines derived from 16C line were resistant to systemic silencing, it is hard to find other explanation than that provided by the authors.

There are just few points that should be added or corrected. Otherwise I regard the study suitable for publication in PLOS One.

Minor comments:

Line 188: Please, provide details on the applied dsRNA (sequences, presence of any modification, method of hybridization of complementary strands, provider of the chemical synthesis).

Line: 152, 184, 363: Please, correct the order, labelling and access to supplementary data. The supplementary files should be referred in the ascending order (S1, S2, S3).

Line 452-454: The sentence is duplicated.

Line 258: Please, indicate whether the double-copy insertion lines used for experiments was homozygous for both integration sites (its sister lines hemizygous for one or two insertions would be a good control in subsequent experiments).

Fig 1 legend: Please correct the sentence “Tissue was collected from the first two true leaves were untreated seedling.”

Fig. 5: Please, provide data for one hemizygous F1 line also in panel B.

If I understand well, the raw data obtained in the study should be publically available, but I did not find a link to repository of the siRNA seq data.

The quality of Figs was really poor in the provided PDF file. It should be checked next time. It is necessary to provide better resolution for publication!!!

Additional suggestion for the authors’ (editor’s) consideration:

1) It is my feeling that the title might be more attractive if switched to "Systemic GFP silencing in Nicotiana benthamiana is associated with high transgene expression"

2) The terminology (e.g. “silencing in plants using topical dsRNA“) and some result details described in the abstract might be not fully understandable/attractive for readers, who are not sufficiently familiar with the issues.¨

3) Transitive siRNA levels were higher in hemizygous lines compared to the double copy line, so it seems that the expression level in distal leaves is more important compared to the expression level in the site of infiltration. Grafting experiments might be helpful.

4) The order of analyzed lines might be the same in different figures

6. PLOS authors have the option to publish the peer review history of their article (what does this mean?). If published, this will include your full peer review and any attached files.

Reviewer #1: **Yes: **Michael Wassenegger

Reviewer #2: No

---

## [Author Response · Author response to Decision Letter 0]

11 Feb 2021

Our responses and changes are detailed following each bulleted reviewer comment below. Line number reference the “Manscript.docx” file 

Reviewer #1

Major points:

• We conclude that high transgene expression level is a key enabler of systemic transgene silencing in N. benthamiana.

In addition to the suggested high transgene expression level, the efficiency of the silencing trigger may have an impact on the initiation of systemic silencing. However, this issue has not been addressed in the present study. Thus, the conclusion that the transgene expression level is a key enabler of systemic transgene silencing should be taken with care. Although the CD-associated 22nt siRNA induced systemic silencing in the homozygous 16C line application of CD:dsRNAs may have an impact on the initiation of systemic silencing. Thus, this reviewer strongly recommends to conduct agroinfiltration experiments using hemizygous 16C plants. Infiltration of an efficient GFP silencing inducer, e.g. GFP hairpin RNA transgene construct, will provide additional evidence that hemizygous 16C plants are indeed unable to develop systemic silencing.

We agree our conclusion statement was too broadly worded in the abstract and closing paragraph. We narrowed our conclusion to be specific for the topically-induced technology described. Lines 43, 127-129, 417-419, 445-447.

The point of efficacy of silencing trigger is a valid one and we spent a good deal of resources addressing it in our research. The GFP dsRNA used in this study was the most efficient dsRNA targeting GFP we identified through our screening efforts. We added a section of text describing the screening process, included references to relevant publications with more detail, and included the sequences in the supplemental data sections. Line s173-179

Testing agroinfiltration is an excellent suggestion to address the larger question of whether hemizygous lines were capable of systemic silencing using a more efficient silencing inducer. However, we believe this is beyond the scope of the specific questions we were addressing in this research. We sought to examine the role of partial transposase integration previously reported, examine the role of 5’ transitivity, and understand if expression levels were related to topically-induced systemic GFP silencing. Agroinfiltration was a tool we frequently used, especially when the project was initiated, but while this technique was valuable in providing foundational data, it was not a technology we deemed suitable for agricultural use. Thus, our focus was to develop the carbon dot delivery system and understand how to enable systemic silencing using this agriculturally-relevant system. We narrowed our conclusion statements and added text discussing agroinfiltration at Line 432-435

Beyond the system question, the data in Fig. 4 address the point Reviewer #1 makes concerning efficiency of the local GFP silencing induction. In these experiments, we applied dsRNA to each hemizygous line once or twice, separated by a few days to allow new growth to emerge in the latter case. The result was a 4-6x increase in the local GFP silencing but no impact was observed in systemic silencing. We discuss this result in Line 421-439

• sRNA profiles: sense and antisense of 21, 22 an 24nt siRNAs should be presented. The profiles of 19-25nt sRNAs may be included in the Supporting Information. Sense sRNA profiles also contain GFP degradation products. Thus, it is important to separately analyze sense and antisense sRNA reads. A focus on 21, 22 and 24nt siRNAs will accentuate the number and distribution of the putatively functional sRNAs that are predominantly DCL products.

In particular, in the “systemic leaves” of NT_W22241793 and NT_W22241802 lines, only the presence of antisense sRNAs will indicate that the accumulating sRNAs derived from the silencing machinery. In this context, it is also important to show CHL-H-specific antisense siRNA (21, 22, ant 24nt) accumulation.

These data are now included in the revised manuscript. We updated all of our figures to present sense and antisense reads separately using the +/- convention to indicate sense/antisense. We left the size distribution histograms in the figures, but we can move to the supplemental section if you feel this would improve the figures.

• The authors should also comment on the observation that “5’ transitive” sRNAs of the CHL-H transcripts are missing. Why no 5’ mRNA degradation products with sense orientation are detected? Finally, in Fig. S1, separate presentation of sense and antisense sRNAs will probably demonstrate that the sRNAs only correspond to GFP-specific degradation products (only sense sRNAs) and not to DCL products. This reviewer predicts that the number of antisense sRNAs, if detectable at all, will be extremely low. In Fig. S1, the size distribution of the sRNAs would make sense. It will show if 21nt sRNAs are predominantly accumulating or if 19 to 25 nt sRNAs are almost equally accumulating. Equal distribution of 19 to 25 nt sRNAs would argue for a mRNA degradation-based origin.

We added a discussion of the 5’/3’ transitivity for CHL-H at Line 439-443. This phenomenon needs further study to understand the details, but we believe the ability to topically deliver specific 22nt dsRNAs using the carbon dot technology described and referenced is a key enabler of these studies in plants.

A small number of sRNA 5’ of the CHL-H were detected but obscured in the original figure. These are visible in the revised figure.

Concerning Fig S1. We updated the figure to parse sense and antisense reads and added a size distribution graphic. Low numbers of both sense and antisense small RNA were detected in the untreated transgenic lines. The reviewer was correct about the size distribution of the small RNAs and we agree with the assessment that the equal size distribution indicated mRNA degradation rather than DCL products.

Minor points:

• Page 11, line 86: Using a grafting approach, the authors showed that DCL2 was required in distal tissue to respond to mobile silencing signal but not required in the initiating tissue to produce the signal. This issue is controversially discussed. Taochy et al. (2017) reported that DCL2 is required in both the source rootstock and the recipient shoot tissue for efficient RDR6-dependent systemic PTGS.

We cited the Taochy paper and added statement that species difference can contributes DCL requirements for systemic silencing Line 84-86 

• Page 15, line 192: The solution was applied to the leaf surfaces using an Iwata HP-M1 handheld airbrush sprayer. Please, indicate the applied pressure. 

We added pressure and distance from leaf surface details to the methods section Line186-187 

• Where the authors purchase from Silwet S279? I only know BreakThru S279 and Silwet L-77. 

Corrected to read BreakThru S279 Line 185

Reviewer #2

Minor comments:

• Line 188: Please, provide details on the applied dsRNA (sequences, presence of any modification, method of hybridization of complementary strands, provider of the chemical synthesis). 

Added a paragraph in materials and methods section describing these details Lines 173-179

• Line 452-454: The sentence is duplicated. 

We deleted the duplication.

• Line 258: Please, indicate whether the double-copy insertion lines used for experiments was homozygous for both integration sites (its sister lines hemizygous for one or two insertions would be a good control in subsequent experiments). 

All lines were confirmed homozygous in the R2 before using in experiment. Added and sentence to clarify at Line 260-261. Also addressed in the methods section Line 146-153-171.

• Fig 1 legend: Please correct the sentence “Tissue was collected from the first two true leaves were untreated seedling.” 

Corrected as indicated Line 243

• Fig. 5: Please, provide data for one hemizygous F1 line also in panel B

Revised Fig 5 to include the small RNA distributions in ‘F1-2’ hemizygous line in panel B.

• If I understand well, the raw data obtained in the study should be publically available, but I did not find a link to repository of the siRNA seq data. 

Sequencing data files are now available in the NCBI SRA database under BioProject ID PRJNA695190

• The quality of Figs was really poor in the provided PDF file. It should be checked next time. It is necessary to provide better resolution for publication!!! 

Corrected using PACE tool

Additional suggestion for the authors’ (editor’s) consideration:

1) It is my feeling that the title might be more attractive if switched to "Systemic GFP silencing in Nicotiana benthamiana is associated with high transgene expression" 

Corrected as indicated

2) The terminology (e.g. “silencing in plants using topical dsRNA“) and some result details described in the abstract might be not fully understandable/attractive for readers, who are not sufficiently familiar with the issues.¨

We noted this suggestion and revised the abstract for clarity. 

3) Transitive siRNA levels were higher in hemizygous lines compared to the double copy line, so it seems that the expression level in distal leaves is more important compared to the expression level in the site of infiltration. Grafting experiments might be helpful. 

Grafting experiments would be useful to for addressing the point the Reviewer makes, but we view this as a large question that is better addressed in a separate line of research. 

The integration locus might impact transitive sRNA levels in the hemizygous versus double copy line as well. We added a statement to this effect at Line 421 459

4) The order of analyzed lines might be the same in different figures 

This is a good suggestion that helps with readability. We revised all figures to match the order of the lines in Fig 1

---

## [Decision Letter · Decision Letter 1]

22 Feb 2021

Systemic GFP silencing is associated with high transgene expression in Nicotiana benthamiana

PONE-D-20-40871R1

Dear Dr. Hendrix,

We’re pleased to inform you that your manuscript has been judged scientifically suitable for publication and will be formally accepted for publication once it meets all outstanding technical requirements.

Kind regards,

Keith R. Davis

Academic Editor

PLOS ONE

Additional Editor Comments (optional):

Reviewers' comments:

Reviewer's Responses to Questions

**Comments to the Author**

1. If the authors have adequately addressed your comments raised in a previous round of review and you feel that this manuscript is now acceptable for publication, you may indicate that here to bypass the “Comments to the Author” section, enter your conflict of interest statement in the “Confidential to Editor” section, and submit your "Accept" recommendation.

Reviewer #1: All comments have been addressed

Reviewer #2: All comments have been addressed

2. Is the manuscript technically sound, and do the data support the conclusions?

Reviewer #1: Yes

Reviewer #2: Yes

3. Has the statistical analysis been performed appropriately and rigorously? 

Reviewer #1: Yes

Reviewer #2: Yes

4. Have the authors made all data underlying the findings in their manuscript fully available?

Reviewer #1: Yes

Reviewer #2: Yes

5. Is the manuscript presented in an intelligible fashion and written in standard English?

Reviewer #1: Yes

Reviewer #2: Yes

6. Review Comments to the Author

Reviewer #1: (No Response)

Reviewer #2: (No Response)

7. PLOS authors have the option to publish the peer review history of their article (what does this mean?). If published, this will include your full peer review and any attached files.

Reviewer #1: **Yes: **Michael Wassenegger

Reviewer #2: No

---

## [Editor Report · Acceptance letter]

5 Mar 2021

PONE-D-20-40871R1 

Systemic *GFP* silencing is associated with high transgene expression in *Nicotiana benthamiana*

Dear Dr. Hendrix:

I'm pleased to inform you that your manuscript has been deemed suitable for publication in PLOS ONE. Congratulations! Your manuscript is now with our production department. 

Kind regards, 

on behalf of

Dr. Keith R. Davis 

Academic Editor

PLOS ONE